# Modeling the evolution of COVID-19 via compartmental and particle-based approaches: Application to the Cyprus case

**Constantia Alexandrou**[1,2☉], **Vangelis Harmandaris**[2,3,4☉], **Anastasios Irakleous**[1,2☉], **Giannis Koutsou**[2☉]*, **Nikos Savva**[2☉]

**1** Department of Physics, University of Cyprus, Nicosia, Cyprus, **2** Computation-based Science and Technology Research Center, The Cyprus Institute, Nicosia, Cyprus, **3** Department of Mathematics and Applied Mathematics, University of Crete, Heraklion, Greece, **4** Institute of Applied and Computational Mathematics, Heraklion, Greece

☉ All authors contributed equally to this work and they thus appear in alphabetical order.
* g.koutsou@cyi.ac.cy

**Data Availability Statement:** The software and data for producing all plots in the manuscript are accessible on GitHub (https://github.com/g-koutsou/2008.03165) and in the Open Science

## Abstract

We present two different approaches for modeling the spread of the COVID-19 pandemic. Both approaches are based on the population classes susceptible, exposed, infectious, quarantined, and recovered and allow for an arbitrary number of subgroups with different infection rates and different levels of testing. The first model is derived from a set of ordinary differential equations that incorporates the rates at which population transitions take place among classes. The other is a particle model, which is a specific case of crowd simulation model, in which the disease is transmitted through particle collisions and infection rates are varied by adjusting the particle velocities. The parameters of these two models are tuned using information on COVID-19 from the literature and country-specific data, including the effect of restrictions as they were imposed and lifted. We demonstrate the applicability of both models using data from Cyprus, for which we find that both models yield very similar results, giving confidence in the predictions.

## 1 Introduction

The COVID-19 pandemic is a new disease and there is as yet not enough understanding on its future evolution. Since medical interventions, such as vaccines or antiviral treatments, are not available yet, non-medical interventions are being implemented to contain the disease. In a number of countries, including Cyprus, the imposed restrictions have helped slow down the spread of the disease. Cyprus, being an island country, managed to limit the spread of the disease, by imposing restrictions on air travel and shutting down large parts of the economy, ultimately achieving 2.2 deaths per 100,000 population, a death rate comparable to that of Greece and Malta [1, 2]. However, as restrictions are being lifted, it is important to know how the disease in each country will evolve. Reliable predictions will help policy-makers to formulate appropriate intervention strategies, while taking into account also economic and social factors.

Framework repository (DOI: 10.17605/OSF.IO/BP79H).

**Funding:** AI is supported by the project "Modeling of the COVID-19 pandemic for Cyprus", contract number CONCEPT-COVID/0420/0011, funded by the Cyprus Research and Innovation Foundation: https://www.research.org.cy VH and NS are supported by project "SimEA", funded by the European Union's Horizon 2020 research and innovation programme under grant agreement No 810660: https://ec.europa.eu/programmes/horizon2020/en The funders had no role in study design, data collection and analysis, decision to publish, or preparation of the manuscript.

**Competing interests:** The authors have declared that no competing interests exist.

Mathematical models and numerical simulation can be used as a decision support tool to assist policy-makers, by forecasting the spread of the disease as a function of the lifting of restrictions as well as on the level of testing and contact tracing [3–12]. Since the epidemic spreading is a complex process depending to a large extent both on the behavior of the virus [4, 5] and human interactions [6, 7], the purpose of this work is to provide such predictions for a number of *scenarios* using two models. While these models are applicable to any country, they are calibrated for the specific case of Cyprus for which such forecasting is not available.

Since the outbreak of COVID-19, a number of mathematical epidemiological models have been used to predict the spread of the disease in a number of countries (e.g. Refs. [3, 7–10]) and regions (e.g., Refs. [11, 12]). Typically, the predictions are made within a single type of model. In this work, we use two models, each relying on different methodologies, to cross-check predictions.

1. *Compartmental models based on ordinary differential equations (ODEs).* These models describe how portions of the population transition among classes or compartments via ODEs. For example, the classical SIR model describes the evolution of three compartments, the susceptible, infected, and removed [13], with two parameters that model the rates at which the population transfers from one compartment to the other. In the present study, our objective is to derive a model that is suitable for countries like Cyprus, where data typically used for modeling COVID-19, such as of hospitalizations, intubation, and deaths, are too small for a meaningful data-driven analysis. We therefore use a time-dependent infection rate and detection rate, with which we are able to capture the reported cases in Cyprus without needing to revert to models with a large number of parameters relying on data that are scarce or not available.

2. *Particle model.* Such models belong to the broader class of crowd simulation/agent-based models, which track the evolution of the interactions of agents in time and space. Here, particle dynamics and interactions are used as a proxy of human social interactions, allowing the transmission of the disease through particle collisions, naturally introducing an element of randomness in the system. This approach is reminiscent of the equivalent of network models and stochastic-branching processes, which is often used to model disease outbreaks. Such models are computationally demanding and an efficient code will be employed in order to simulate the whole population of Cyprus residing in cities, where most of the COVID-19 cases are registered.

The main goal of the study is to examine the forecasting potential of the above models for the short-term evolution of COVID-19, under various conditions related to imposing or lifting measures. We demonstrate that with a relatively small set of parameters both models describe quite accurately the existing data and yield very similar predictions for the evolution of the disease when the same assumptions are made for the measures imposed and lifted. More importantly, we examine whether such models predict the number of positive COVID-19 cases for small countries like Cyprus assuming a number of different scenarios for the evolution of the infection rate, as the economy reopens and in relation to the ability of the health care services to detect and isolate future cases through contact tracing and aggressive testing. One of the key highlights of this study is that, although these models are based on different approaches, they both yield consistent predictions within their corresponding uncertainties. This leads to confidence on the model forecasts on one hand, but also allows us to attribute different errors to the predictions, since the particle model exhibits stochastic errors, related to how the pandemic was seeded within the population, as well as uncertainties in the parameters, such as the velocities of the particles, while the compartmental model exhibits modeling uncertainty, based on

the uncertainties of the fitted parameters. This study also demonstrates the ability of these models to capture features of the pandemic, such as the infection rate, when relatively small numbers of cases are involved and for which statistical models yield results with very large uncertainties. Their reliability, on the other hand, is also subject to changes in the way the population collectively behaves during the forecasting period.

The structure of the paper is as follows: In Sec. 2 we introduce the two models and describe how we tune the parameters, adjusting them to the specific case of Cyprus. In Sec. 3 we discuss the results of the models and in Sec. 4 we give our conclusions and future plans.

## 2 Description of the models

We describe here the two models and how their parameters are tuned. There is presently a plethora of models of varying degrees of complexity and sophistication, depending on the wealth of data available. For example, for larger populations, where naturally the death toll is higher, modeling approaches may be effectively used to capture the evolution of the number of deceased [14–16]. For smaller countries, however, like Cyprus with a small number of deaths, the applicability of e.g. statistical models that focus on extracting multitude of parameters from the recorded deaths [3] may be questionable. Here we propose an approach that uses a minimal set of fitting parameters. To model COVID-19 evolution one needs to make certain assumptions. In our modeling we use some common features that are also being applied in a number of other studies that are currently being applied to COVID-19 [3, 7–10]. Namely, we consider that

i. The disease consists of two phases: (a) an exposed phase during which an individual contracts the disease but has no symptoms and does not transmit the disease to others; (b) an infectious phase, during which individuals transmit the disease to susceptible members of the population.

ii. A portion of the exposed individuals who ultimately become infectious is detected. We take this portion to be a function of time depending on the level of screening and testing being performed.

iii. The detected cases are assumed to be immediately quarantined, so that the spread of the virus is attributed solely to the undetected cases.

iv. The population is divided in five classes; the susceptible, exposed, quarantined, infectious, and removed; i.e. an SEIQR type of model.

v. The removed class includes individuals who (a) recover or (b) die due to the disease and we do not differentiate among these subclasses. Deaths can be modeled by inputing the death rate, which also depends on the capacity of healthcare systems to accommodate the needs of those who become critically ill. While in the future such an analysis may be possible, at the time of writing, there are not enough data available for the case of Cyprus to reliably extract a death rate. In addition, the quarantined class is modeled in the same way as the removed class, but tracked independently as a function of time

vi. The average duration of the exposed latent period is constant.

vii. The average duration of the infectious period is constant.

viii. A key feature of our modeling is that we take the rate of infection/transmission to be a function of time, reflecting the measures implemented by the government in regards to social distancing, limitations of travel, and suspending parts of the economy.

For the infectious period, we rely on recent data that estimate it to be 7-12 days. [17]. We therefore fix the infectious period ($\tau_i$) to 10 days and the exposed period ($\tau_e$) to 2 days.

These modeling features are kept the same for both our models. Further extensions of the models could include, for example, assigning a probability of the quarantined population to infect, a delay between the time an infectious individual is quarantined, or defining additional groups within the classes which would however introduce more parameters. While such extensions of the models may be considered in the future as more data become available, we opt to evaluate here our models with the least possible parameters, which as will be seen in Sec. 3 are sufficient to describe the current data.

## 2.1 Compartmental model

Extending the original SIR model of Kermack and McKendrick [13] to include an Exposed and a Quarantined class, the following coupled system of ODEs arises, which is in alignment with the assumptions introduced earlier:

$$\frac{\mathrm{d}S}{\mathrm{d}t} = -\beta \frac{SI}{N}, \tag{1a}$$

$$\frac{\mathrm{d}E}{\mathrm{d}t} = \beta \frac{SI}{N} - \frac{E}{\tau_e}, \tag{1b}$$

$$\frac{\mathrm{d}I}{\mathrm{d}t} = r\frac{E}{\tau_e} - \frac{I}{\tau_i}, \tag{1c}$$

$$\frac{\mathrm{d}Q}{\mathrm{d}t} = (1-r)\frac{E}{\tau_e} - \frac{Q}{\tau_i}, \tag{1d}$$

$$\frac{\mathrm{d}R}{\mathrm{d}t} = \frac{I+Q}{\tau_i}, \tag{1e}$$

where $S(t)$, $E(t)$, $I(t)$, $Q(t)$, and $R(t)$ capture, respectively, the evolution of the susceptible, exposed, infected, quarantined, and removed classes of the population as a function of time $t$. We will refer to this extended model as the SEIQR model. The parameter $\beta > 0$ corresponds to the infection rate in inverse time units, which is a measure of the average number of contacts an infective individual makes in a wholly susceptible population that may lead to an infection per unit time, $\tau_i > 0$ and $\tau_e > 0$ are the average times an individual remains in the infective and exposed classes respectively. The portion of undetected cases who later infect others is given by $r$ and $N$ is the total population (assuming no vital dynamics, namely births or deaths due to other causes).

Central to the standard SIR and SEIQR models is the modeling of the rates at which individuals transition between population classes. Both the SIR and SEIQR models assume that the times for which individuals remain in the exposed and infectious states are exponentially distributed random variables [18], which implies that the chance of an individual moving out of the exposed and recovered classes is independent of the time they entered the particular class. This leads to dispersed timescales, which manifests itself, for example, in unrealistically long recovery times for the number of individuals that got infected towards the end of an epidemic and in overoptimistic predictions of the levels of control required to contain the epidemic [19]. A more general approach assigns arbitrary probability distributions for the recovery and

latent times. This yields a system of integral–differential equations of the form

$$E(t) = E(0)P_E(t) + \int_0^t \beta \frac{S(\tau)I(\tau)}{N} P_E(t-\tau)\, \mathrm{d}\tau, \tag{2a}$$

$$I(t) = I(0)P_I(t) + r \int_0^t \left( \beta \frac{S(\tau)I(\tau)}{N} - \frac{\mathrm{d}}{\mathrm{d}\tau} E(\tau) \right) P_I(t-\tau)\, \mathrm{d}\tau, \tag{2b}$$

$$Q(t) = Q(0)P_I(t) + (1-r) \int_0^t \left( \beta \frac{S(\tau)I(\tau)}{N} - \frac{\mathrm{d}}{\mathrm{d}\tau} E(\tau) \right) P_I(t-\tau)\, \mathrm{d}\tau, \tag{2c}$$

$$R(t) = (Q(0) + I(0))(1 - P_I(t)) - \int_0^t \left( \beta \frac{S(\tau)I(\tau)}{N} - \frac{\mathrm{d}E(\tau)}{\mathrm{d}\tau} \right) (1 - P_I(t-\tau))\, \mathrm{d}\tau, \tag{2d}$$

$$S(t) + E(t) + I(t) + R(t) = N, \tag{2e}$$

where $P_I(t)$ and $P_E(t)$ are non-increasing functions that correspond to the probabilities of remaining infectious or quarantined and exposed $t$ units after becoming infectious or quarantined and exposed, respectively with $P_I(0) = P_E(0) = 1$ (see [18] for a related model). In each of Eqs (2a)–(2c), the first terms correspond to the respective initial populations in a class that remain in the same class after $t$ time units, whereas the second terms correspond to the sum of individuals who become members of a class within the time interval $[0, t]$. Eq (2d) captures the transfer of the infectious and quarantined classes to the removed classes, whereas combining Eqs (2a)–(2e) yields Eq (1a).

As noted in earlier works (e.g. Ref. [18]), letting $P_E(t) = \mathrm{e}^{-t/\tau_e}$ and $P_I(t) = \mathrm{e}^{-t/\tau_i}$ reduces Eqs (2) to (1). Although it has been argued that more general multi-stage gamma-distributed latent and infectious periods may be more appropriate, see e.g. [19–21], precise knowledge of $P_E(t)$ and $P_I(t)$ is neither known nor expected to have an appreciable qualitative effect. Here we chose the latent and infectious periods to be of fixed length as means to alleviate the aforementioned issues if these are exponentially distributed, see also Ref. [22]. By doing so, we manage to preserve the simplicity of the model, allowing us also to obtain a discrete set of equations. Hence $P_E(t)$ and $P_I(t)$ are assumed to be of the form

$$P_E(t) = \begin{cases} 1, & 0 \le t < \tau_e \\ 0, & t \ge \tau_e \end{cases}, \qquad P_I(t) = \begin{cases} 1, & 0 \le t < \tau_i \\ 0, & t \ge \tau_i \end{cases}. \tag{3a, b}$$

Considering this time-dependence, we are able to deduce the following system of delay

differential equations

$$\frac{dS}{dt} = -X(t), \tag{4a}$$

$$\frac{dE}{dt} = X(t) - X(t - \tau_e), \tag{4b}$$

$$\frac{dI}{dt} = rX(t - \tau_e) - rX(t - \tau_i - \tau_e) - I_0\delta(t - \tau_i), \tag{4c}$$

$$\frac{dQ}{dt} = (1 - r)X(t - \tau_e) - (1 - r)X(t - \tau_i - \tau_e) - Q_0\delta(t - \tau_i) \tag{4d}$$

$$\frac{dR}{dt} = X(t - \tau_i - \tau_e) + I_0\delta(t - \tau_i) + Q_0\delta(t - \tau_i), \tag{4e}$$

where, for $t > 0$,

$$X(t) = \beta \frac{S(t)I(t)}{N} \tag{4f}$$

when $t > 0$ and $\delta(t)$ being the Dirac delta function.

The system of Eq (4) is further extended in the present study by incorporating the features given by items ii), iii) and vi) above, achieved by introducing a time-dependent function $r(t)$ which corresponds to the portion of undetected cases.

Furthermore, rather than solving the delay differential equations that may require specialized techniques, we opt to convert the system of Eq (4) to a discrete difference equation, using a time step of 1, so that they become:

$$S(t + 1) = S(t) - X(t), \tag{5a}$$

$$E(t + 1) = E(t) + X(t) - X(t - \tau_e), \tag{5b}$$

$$I(t + 1) = I(t) + r(t - \tau_e)X(t - \tau_e) - r(t - \tau_i - \tau_e)X(t - \tau_i - \tau_e) - I_0\delta_{t,\tau_i}, \tag{5c}$$

$$\begin{aligned} Q(t + 1) &= Q(t) + (1 - r(t - \tau_e))X(t - \tau_e) \\ &\quad - (1 - r(t - \tau_i - \tau_e))X(t - \tau_e - \tau_i) - Q_0\delta_{t,\tau_i} \end{aligned} \tag{5d}$$

$$R(t + 1) = R(t) + X(t - \tau_i - \tau_e) + (Q_0 + I_0)\delta_{t,\tau_i}, \tag{5e}$$

where $t$ represents discrete time in days and $\delta$ is the Kronecker delta. The choice of the form of $r(t)$ will be discussed in Sec. 3. The functional forms of Eq (5) are to be solved with the appropriate initial conditions. We have confirmed numerically that such an approach does not compromise the overall quantitative agreement with the solutions to the original system of delay differential equations. Note that the equations for $E(t)$, $Q(t)$ and $R(t)$ may be decoupled from the system, as they can be fully specified independently once the evolution of $X(t)$, $I(t)$, and $S(t)$ is determined. Hence, it suffices to keep track of the movement of individuals across the susceptible and infective classes as well as the number of exposed individuals at each day. Initially, we take $I(0) = r(0)c_0/(1 - r(0))$, the actual undetected cases who are assumed to be responsible for initiating the epidemic outbreak as derived from the initial confirmed cases $c_0$.

The exposed individuals at $t = 0$ are those who become infectious at $t = \tau_e$ and hence $X(0) = (c_2 - c_1)/(1 - r(0))$, where $c_1$ and $c_2$ are the confirmed cases on days 1 and 2, respectively. Likewise, the exposed individuals one day before the first cases are confirmed at $t = 0$, will be those who will become infected when $t = 1$, so that we may take $X(-1) = (c_1 - c_0)/(1 - r(0))$, assuming for simplicity that $r(0) = r(-1)$. This allows us to determine the susceptibles at $t = 0$ as $S(0) = N - c_2/(1 - r(0))$. Summarizing, we consider $X(t)$ of the form

$$X(t) = \begin{cases} \dfrac{c_1 - c_0}{1 - r(0)}, & t = -1 \\[2mm] \dfrac{c_2 - c_1}{1 - r(0)}, & t = 0 \\[2mm] \dfrac{\beta(t)S(t)I(t)}{N}, & t > 0 \end{cases}, \tag{6}$$

with $\beta(t)$ being time-dependent to reflect governmental measures imposed or lifted. The fitting process is facilitated by the fact that the government imposes or relaxes measures in $M$ stages and at given times $t_1, t_2, \ldots, t_M$. In order to use the fewest possible parameters, we take $\beta(t)$ to be of the form

$$\beta(t) = \frac{1}{2}\left[ b_M + b_0 + \sum_{j=1}^{M} \left( b_j - b_{j-1} \right) \tanh \left( m_j(t - t_j) \right) \right], \tag{7}$$

where $b_0$ and $b_M$ being, respectively, the initial and final transmission rates ($\lim_{t \to \infty} \beta(t) = b_M$), $b_j, j = 1, \ldots, M - 1$ correspond to intermediate transmission rates, and $\tanh(x) = \frac{1 - e^{-2x}}{1 + e^{-2x}}$ is the hyperbolic tangent function. This choice for modeling the time dependence of the infection rate is very flexible, allowing us to cover a broad range of functional time dependence of $\beta(t)$. Namely, the parameters $m_j$ control how smoothly $b_{j-1}$ transitions to $b_j$ so that $1/m_j$ gives an order of magnitude of how long this transition lasts (as $m_j \to \infty$ this transition becomes step-like).

The model parameters are obtained by fitting the data for confirmed cases $c_t$, to our model confirmed cases, given by:

$$C(t) = c_0 + \sum_{\tau = \tau_e - 1}^{t} \left(1 - r(\tau - \tau_e)\right) X(\tau - \tau_e) \tag{8}$$

through a least-squares fit. To fit the initial stages until July 2020 of the Cyprus case for which this study was undertaken, i.e. during the first lock-down and the gradual lifting of measures following it, we consider a two-stage process ($M = 2$) based on the actual announcements by the government, with measures enforced on March 24th, 2020 ($t_1 = 15$) and almost fully lifted on May 21st, 2020 ($t_2 = 73$). This leads to a five parameter fit for $b_0, b_1, b_2, m_1$, and $m_2$. If we were to consider a more extended period of time, then the method can be automated to enable the determination of $M$ and the times $t_j$ by finding the inflection points of $c_t$, i.e. finding the times at which the second derivative changes sign. This is done by first smoothing out $c_t$ with a smoothing spline. Furthermore, for more extended periods of time, $m_1$ and $m_2$ can be set to those obtained by fitting the first lock-down phase, while $m_j$ for $j > 2$ are kept fixed leaving us with an $M + 1$-parameter fit, namely to determine the parameters $b_j, j = 0, \ldots, M$. Therefore, fitting the initial period is still relevant if one wants to use the models for a long period during the pandemic.

The reported as recovered would require the introduction of yet another timescale, since according to the protocols followed, individuals are considered to have recovered after they test negative twice within a period of 24 hours and only after all symptoms are resolved, which leads to a median recovery time of 23 days [23]. The portion of undetected $r(t)$ evolves in a prescribed manner that captures how aggressively testing is performed, and is elaborated in Sec. 3.

Below we present plausible scenarios for the evolution of COVID-19 in Cyprus based on the knowledge of measures as of July 2020. We stress that additional scenarios can be analyzed to reflect new different circumstances as they evolve e.g. an increase of cases from incoming people, etc. The strengths of the models is that they can be adjusted to new measures and human behavior by adjusting the reproduction number. Indeed, the form of the infection rate we have chosen allows our models to capture such changes, as also demonstrated in the addendum in which we include an example of how we can model data that have become available after submission of this work. We should note however, that interpreting and distinguishing between the different factors that contribute to a given change in the reproduction number is non-trivial and beyond the scope of this work.

**2.1.1 Forecasting for various scenarios.** Beyond the available data, which are fitted to determine the five parameters of the model, we forecast the evolution of the epidemic based on different scenarios on how the infection rate $\beta(t)$ will evolve. In one scenario, we choose to divide the three major classes $X(t)$, $S(t)$ and $I(t)$ into two groups each, namely $X_k(t)$, $S_k(t)$ and $I_k(t)$ with $k = 1, 2$ that can be used to model situations where a portion of the population, e.g. people above 65 years old and/or with preexisting conditions, is considered vulnerable to the disease and continues to observe strict social distancing measures compared to the rest who return to work. Hence, initially and up to some time $t = t_*$ the population is assumed to behave uniformly to reflect the lockdown imposed to the whole population. For $t > t_*$, we differentiate into subclasses as follows. We prescribe the $2 \times 2$ contact matrix with constant entries $\beta_{j,k}$, which denotes the average number of infectious contacts made per day by an individual in group $j$ with an individual in group $k$. Since the total number of contacts between group $j$ to $k$ must equal the number of contacts from group $k$ to group $j$, we must have $N_j \beta_{j,k} = N_k \beta_{k,j}$, where $N_1$ and $N_2$ are the corresponding populations of each group with $N = N_1 + N_2$.

Therefore, the exposed, infectious and susceptible for groups 1 and 2 evolve according to

$$X_k(t) = \left(\frac{\beta_{k,1} I_1(t)}{N_1} + \frac{\beta_{k,2} I_2(t)}{N_2}\right) S_k(t) \tag{9a}$$

$$S_k(t+1) = S_k(t) - X_k(t), \tag{9b}$$

$$I_k(t+1) = I_k(t) + r_k(t-\tau_e) X_k(t-\tau_e) - r_k(t-\tau_i-\tau_e) X_k(t-\tau_i-\tau_e), \tag{9c}$$

for $t > t_*$ and $k = 1$ and 2. For $t \leq t_*$ the time histories of $X_k$, $S_k$ and $I_k$ correspond to scalings of $X(t)$, $S(t)$ and $I(t)$ by $N_k/N$, and we also let $r_k(t) = r(t)$. Hence, the confirmed cases for each of the two groups, $C_k$ ($k = 1$ and 2), is given at time $t > t_*$ by

$$C_k(t) = C(t_*)\frac{N_k}{N} + \sum_{\tau=t_*+\tau_e-1}^{t} (1 - r_k(\tau-\tau_e)) X_k(\tau-\tau_e) \tag{10}$$

Whether the two groups evolve differently depends crucially on $r_1(t)$ and $r_2(t)$ and the contact matrix $\beta_{j,k}$. For instance, if we set $r_k(t) = r(t)$ and $\beta_{k,j} = \beta N_j/N$, the collective evolution of the two groups is not distinguishable from a simulation using Eq (5) with the two groups mixed in a single group.

If detailed data on the decomposition of the confirmed cases are available, this two population model, and its generalization to multiple populations, can be used to fit and obtain the entries of the contact matrix as fit parameters. This would allow capturing, for example, heterogeneity between subgroups of the population as regards their infectivity and susceptibility. However, in the case of Cyprus such data are not available and, thus, we fit to a single population model, and limit to using the two population model in the forecasting, to qualitatively assess the effect of different measures being imposed to different subgroups.

## 2.2 Particle model

Within the particle model, disease transmission is modeled by elastic collisions of two-dimensional hard discs. The number of particles per unit time scattered in any direction from a given disc is

$$N_{\text{coll.}} = Dnv_0, \tag{11}$$

where $n$ is the number of discs per unit area with radius $D/2$ and $v_0$ is their velocity. We take $D = 4$ m so that individuals with distance greater than 2 m can not infect others. This fixes the length unit of the system. Then the basic reproduction number is given by

$$\mathcal{R}_0 = N_{\text{coll.}} p \tau_i, \tag{12}$$

where $p$ is the probability that an infectious individual transmits the disease upon collision with another individual. We take $\tau_i$ to be the same average time a single individual is infectious as for the extended SEIQR model.

The work-flow implemented includes using the DynamO [24] particle simulator within our own post-processing scripts to generate a list of elastic particle collisions. This list is then parsed with a given set of parameters, namely initial number of infected, exposed, and quarantined, infection probability for each population subgroup, detection rate as a function of time, and velocities as a function of time. At regular time-steps the total number of susceptible, exposed, infectious, quarantined, and recovered are registered. As in the case of the extended SEIQR model, an exposed individual transitions either to infectious or quarantined based on a time-dependent undetected ratio $r(t)$. Quarantined individuals do not infect and their time evolution is taken to model the reported numbers by the government. Our work-flow, which includes post-processing of the output from DynamO, analyzing the collisions list, and processing and visualizing the data is available online [25].

The particle simulation time units are in a scale that can only be expressed in physical time units *a posteriori*. To determine the time scale $a$, we initialize with one randomly chosen exposed individual and measure the average number of transmissions per individual ($R$) as a function of time, for multiple values of $\tau_i/a$. By convention, we only measure this quantity for individuals that have recovered, which means $R(t) = 0$ for $t < \tau_i + \tau_e$. In the left panel of Fig 1, we show $R(t)$ measured for representative values of $\tau_i$. The measurement is repeated 256 times, randomly varying the initial individual exposed each time, which yields the statistical error in $R(t)$. $\tau_e$ is kept fixed to one fifth of $\tau_i$. From this analysis we see that $R(t)$ is constant for a time period after $t = \tau_i + \tau_e$ long enough to obtain a reliable measurement of $R_0$ for each choice of $\tau_i$. In the right panel of Fig 1 we show the measured $R_0$ as a function of $\tau_i$ and confirm the linear behavior as expected from Eq (12). Demanding that initially $R_0 = 3.5$ and $\tau_i = 10$ days we fix $a$ via a linear fit, which yields $a \approx 2.763$ days per simulation time unit.

After fixing the time scale $a$ we further tune the particle model in order to determine the dependence of $R_0$ on the probability of infection $p$ and velocity scaling factor $u/v_0$. This is shown in Fig 2, for representative values of $p$. This tuning allows us to determine the

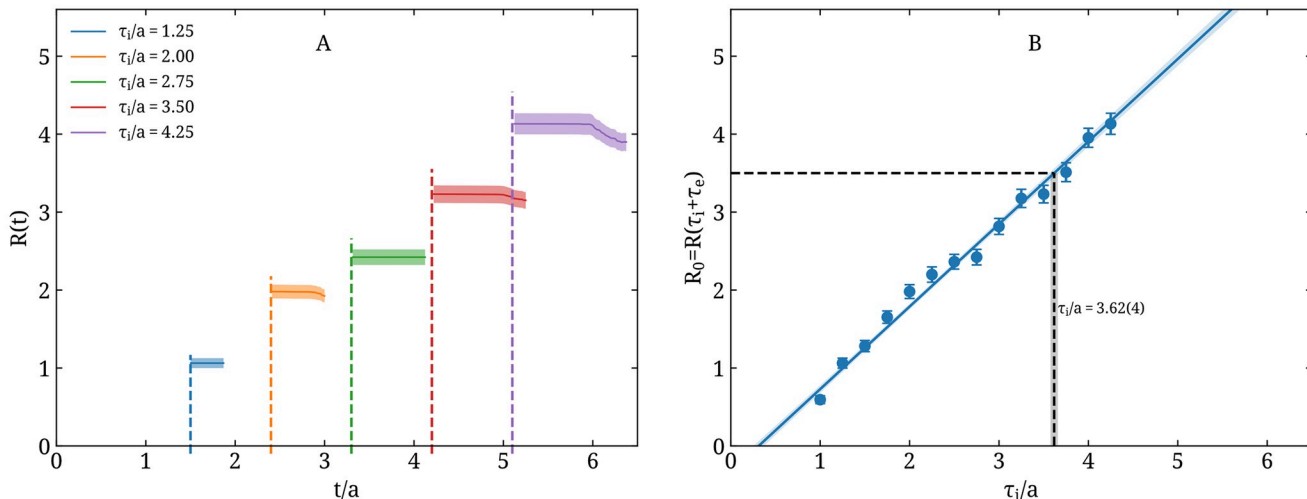

**Fig 1. Setting the particle model time scale.** Left panel (A): The average number of transmissions per individual as a function of time $R(t)$ for representative values of the infectious period in simulation time units ($\tau_i/a$), namely for $\tau_i/a = 1.25$ (blue), 2 (orange), 2.75 (green), 3.5 (red), and 4.25 (purple). The dashed vertical line shows $t = \tau_i + \tau_e$ for each case. The errorband in $R(t)$ is from 256 statistics. Right panel (B): The basic reproduction number $R_0$ obtained as $R(\tau_i + \tau_e)$ from the particle model as a function of the infectious period in simulation time units ($\tau_i/a$). The exposed time $\tau_e$ is fixed to $\tau_e = \tau_i/5$. The blue line and band are the result of a linear fit to $R_0$. The horizontal dashed line shows $R_0 = 3.5$, used to set the scale $a$, via $\tau_i = 10$ days.

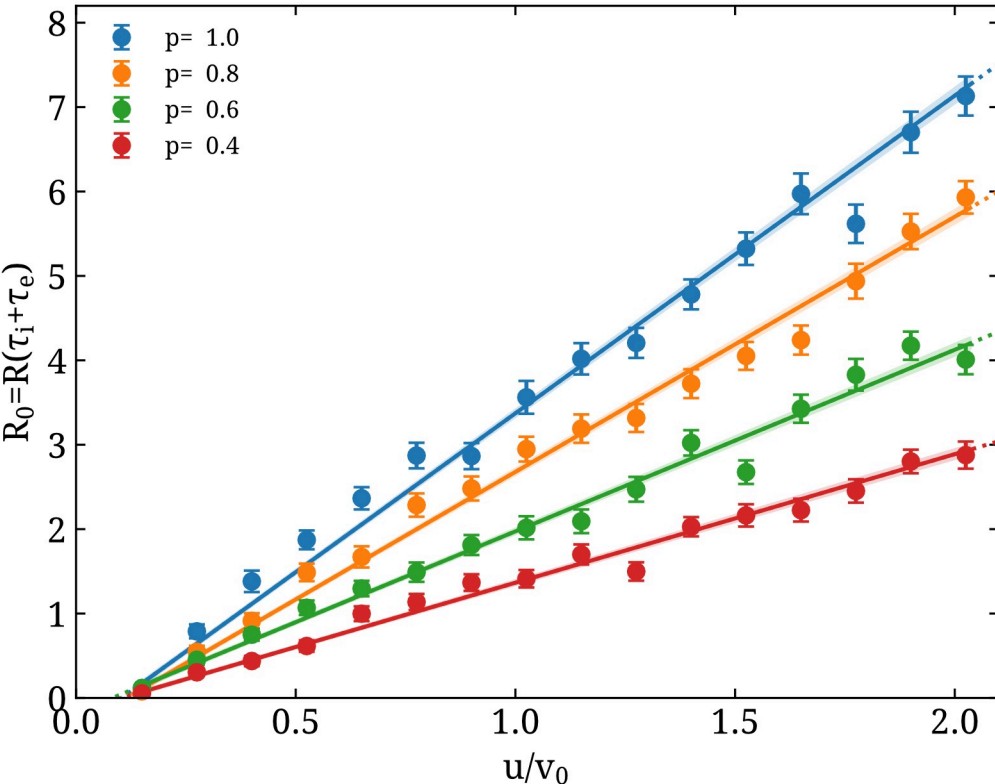

**Fig 2. Particle model tuning.** $R_0$ obtained as a function of the particle velocities $u/v_0$ for representative values of the probability of infection $p$.

combination of $u/v_0$ and $p$ to achieve a desired value of $R_0$. Namely, we use 10 values of $p$ within [0, 1) and fit to the form:

$$\alpha_0 \frac{u}{v_0}p + \alpha_1 \frac{u}{v_0} + \alpha_2 p + \alpha_3 \tag{13}$$

obtaining $\vec{\alpha} = (3.68(4), 0.01(2), -0.37(2), -0.012(8))$.

## 2.3 Determination and comparison of $R(t)$ between models

In the results that follow, we will quote values of the effective reproduction number $R(t)$ obtained from both the extended SEIQR model and the particle model. To facilitate comparison, we define the following:

- With $R^{\text{model}}(t)$, for the extended SEIQR model, we will quote the value of the effective reproduction number of the modeled confirmed cases $C(t)$. This is related to the rate at which individuals transition from Susceptible to Quarantined in Eq (5d), or equivalently the number of new confirmed cases per unit time per infective individual $(1 - r(t))\beta(t)$, multiplied by the period over which the individual is infective, $\tau_i$. It is therefore given by

$$R^{\text{model}}(t) = \beta(t)(1 - r(t))\tau_i, \tag{14}$$

where $\beta(t)$ is of the form of Eq (7) and its parameters will be determined from fits as will be discussed in the next section. For the particle model, we will quote $R^{\text{model}}(t)$ as obtained from Eq (13), where the velocities $u(t)/v_0$ and probabilities $p(t)$ are time-dependent. In particular, for the particle model $R^{\text{model}}(t)$ is given by

$$R^{\text{model}}(t) = \alpha_0 \frac{u(t)}{v_0}p(t) + \alpha_1 \frac{u(t)}{v_0} + \alpha_2 p(t) + \alpha_3, \tag{15}$$

with $\vec{\alpha}$ as determined in Sec. 2.2

- With $R^{\text{integral}}(t)$ we will quote, for both models, an integral definition of $R(t)$ as defined in Ref. [26]. Namely, in this case we take

$$R^{\text{integral}}(t) = \tau_i \frac{\rho(t+1)}{\sum_{i=t-(\tau_i+\tau_e)+1}^{t-\tau_e} \rho(i)} \tag{16}$$

where $\rho(t)$ are the modeled daily new cases at time $t$ as determined by one of our models. With $R^{\text{integral}}_{(data)}(t)$ we indicate this definition used on the actual reported daily new cases of Cyprus, i.e. using $\rho(t) = c_t - c_{t-1}$ in Eq (16). This definition is used since it was shown to account better for the fluctuations shown in the reported cases.

## 3 Results

The two aforementioned models can now be implemented in a country-specific case. In this study we consider the case of Cyprus for which such modeling and forecasting is lacking. Our analysis strategy is to adjust the parameters of the two models to reproduce the available data of Cyprus up to the writing of this manuscript, namely July 31$^{\text{st}}$, 2020 and then predict the evolution until the end of 2020 under different scenarios that reflect the measures and behaviors known at the time of this writing. We then present in an addendum, added after review, the changed situation where incoming population caused an increased in the positive cases and new measures. As will be discussed, both models can be adjusted to take into account the

additional measures and predict correctly the evolution under the updated circumstances. In what follows we will model the initial lock-down phase, up until July 31$^{st}$, 2020, with four different forecasting scenarios as explained below. The software for generating the plots that will be presented in this section is available at Ref. [27], while the data are available at Ref. [28]. Furthermore, the SEIQR model presented here has been adapted to fit data in real-time and is maintained on an online platform [29].

In Fig 3 we show the result of fitting Eq (8) of our model to reported COVID-19 positive cases in Cyprus [23] to obtain the parameters of $\beta(t)$ for the SEIQR model. We validate the

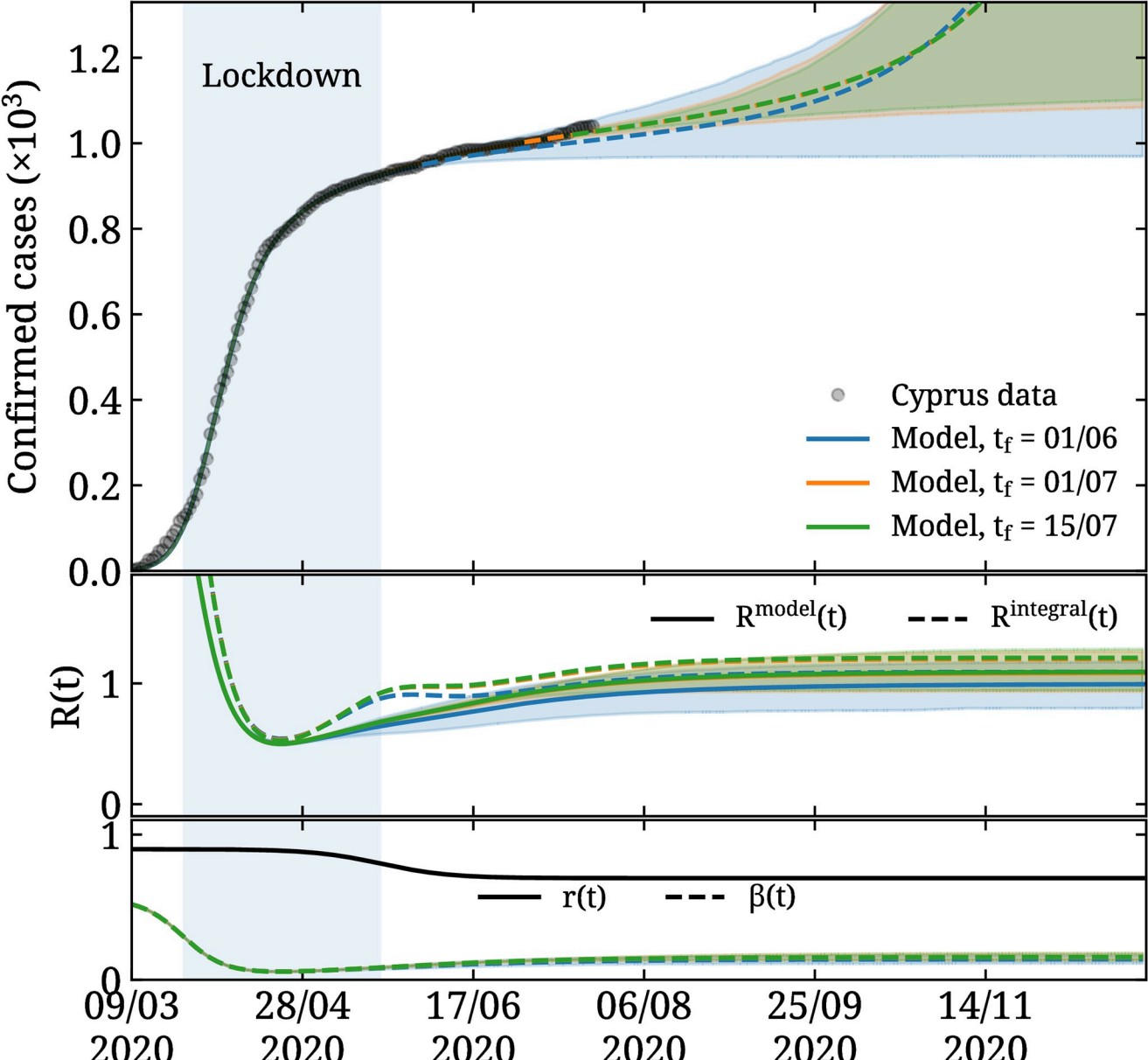

**Fig 3. Fitting the SEIQR model.** In the top panel, we show the reported cases in Cyprus (circles) as a function of time and the result of fitting our SEIQR model. We consider three periods taking $t_f = 84$ (blue curve), 114 (orange curve), and 128 (green curve) that overlap with the data. The blue shaded region shows the interval between March 25$^{th}$ and May 21$^{st}$, used as $t_1$ and $t_2$ in Eq (7). The dashed curves and bands show the predicted mean value and 90% confidence level as the time range being fitted varies by changing $t_f$. The central panel shows $R^{model}$ with the solid curves and $R^{integral}$ with the dashed curves. The bottom panel shows the ratio of undetected $r(t)$.

model by using data up to date $t_f$ and compare with the known cases. We take $t_f$ to be June 1st, July 1st, and July 15th. We sample the fit parameters via Markov Chain Monte Carlo using the Stan library [30], defined with a likelihood function via $e^{-\chi^2/2}$, where $\chi^2$ is given by

$$\chi^2 = \sum_{t=0}^{t_f} \left| \frac{c_t - C(\vec{\theta}, t)}{\sigma} \right|^2, \tag{17}$$

with $c_t$ the cumulative reported cases for Cyprus with $t = 0$ March 9th and $C(\vec{\theta}, t)$ the result from the SEIQR model, namely Eqs (8) or (10). The vector $\vec{\theta}$ contains the parameters to be fitted, namely $\vec{\theta} = (b_0, b_1, b_2, m_1, m_2)$, with $M = 2$ in Eq (7) and $\sigma$ drawn from a normal distribution. We generate 400 independent samples of the parameters obtained using two chains of 2000 iterations, discarding the first 1000 as thermalization iterations and of the remaining 1000 we took every fifth in each chain.

At this point in time, the number of actual cases as compared to the reported are not known. For example, an early study analyzing testing data from Iceland put this number between five to ten times. There, intensive testing was carried out early on, obtained from 1st to 4th of April [31]. A study in the US, analyzing data obtained from 23rd March to 12th May, estimated this number to be between six to 24 times [32].

In Cyprus, we observe an intensification of testing and tracing since May [23] and a low death rate [2], which indicates low prevalence of the virus. We take the portion of undetected cases $r(t)$, to gradually decrease from one detected in 10 infected to one in three by using the functional form for the portion of undetected cases $r(t)$ to be

$$r(t) = \frac{r_1 + r_0}{2} + \frac{r_1 - r_0}{2} \tanh\left( m_r(t - t_r) \right), \tag{18}$$

with $r_0 = 0.9$, $r_1 = 0.7$, $m_r = 0.05$, and $t_r = 73$ or May 21st, 2020. This choice is motivated by the evolution of daily tests and positivity rate, shown in Fig 4, which stabilize around May to about 2000 tests per day at a positivity rate of about 0.1% [33].

As can be seen, we obtain consistent results and good prediction of the available data when fitting until June 1st i.e. when omitting up to two months of the most recent data from the fits. From the central panel, we see that the estimates for $R(t)$ overlap when varying $t_f$, which confirms that our fits yield consistent parameters for the choices considered. The parameters obtained for each fit range are tabulated in Table 1.

As mentioned, we use $r_0 = 0.9$ and $r_1 = 0.7$ motivated by the ramping up of daily testing in Cyprus during May 2020. We note that our fits are robust to small changes to $r_0$ and $r_1$ as long as their ratio is maintained and excluding extrema such as $r_0 \simeq 1$ and $r_1 \simeq 0$. In Fig 5 we compare two choices for $r_0$ and $r_1$ in addition to those used in Fig 3 for the case of $t_f = 128$. We see that indeed the evolution of the pandemic up to the end of October 2020 is within errors for all three choices, and thus that our predictions are not sensitive to the precise inputs for $r_0$ and $r_1$. For the remainder of this paper we will use the choice $r_0 = 0.9$ and $r_1 = 0.7$.

Having validated our extended SEIQR model by predicting the reported cases over the known period by varying $t_f$, we make forecasting using four plausible scenarios:

A) In scenario A, we forecast the evolution of the pandemic assuming infection and testing rates remain unchanged, i.e. we take $\beta(t)$ and $r(t)$ to remain the same after the fitted period.

B) In scenario B, we investigate the case where $r(t)$ asymptotically reaches 50% by mid August. This reflects a scenario in which more aggressive testing and contact tracing is performed.

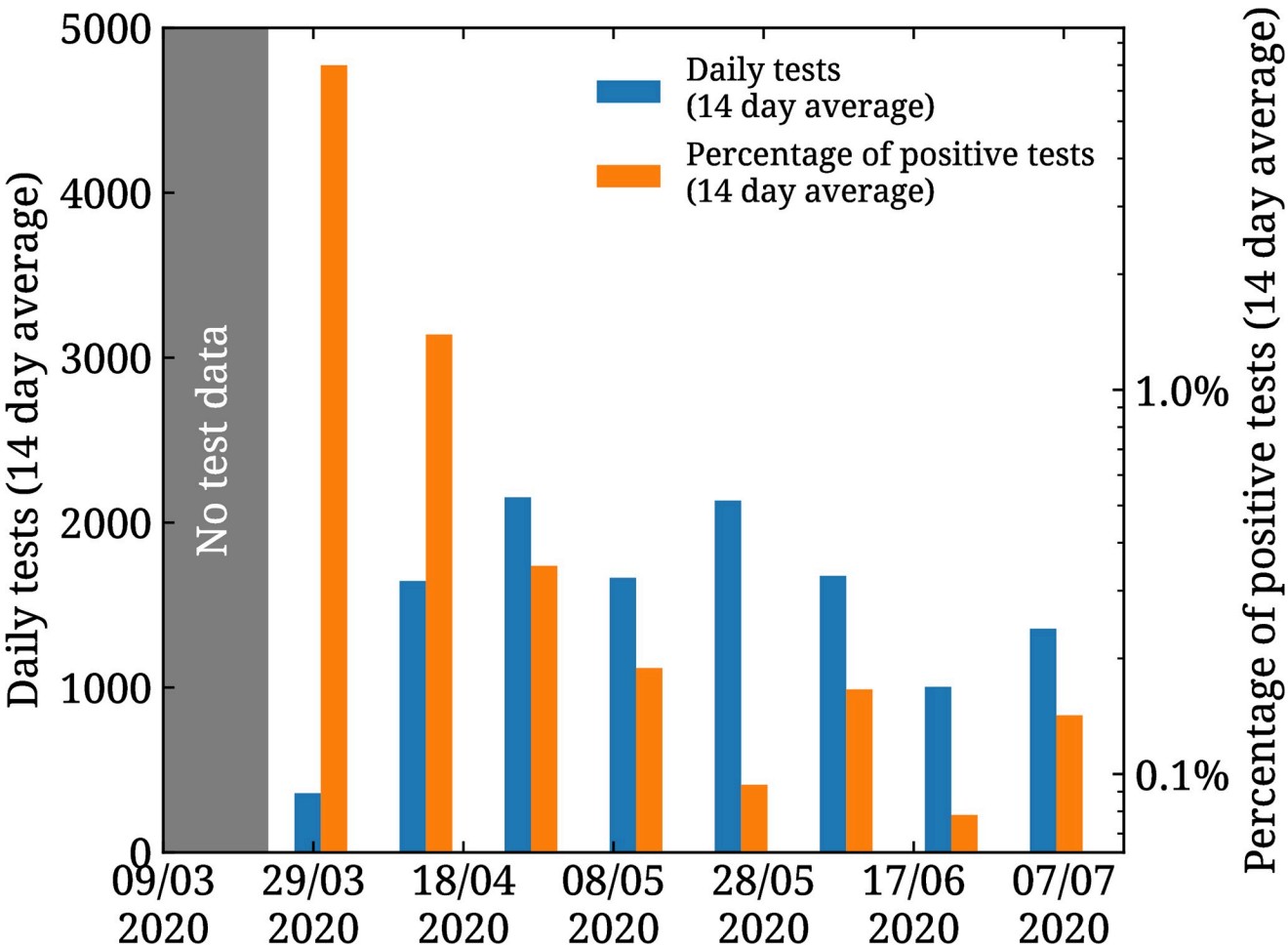

**Fig 4. Daily tests and positivity rate in Cyprus.** 14-day average of the daily tests (blue bars) and of the percentage of daily tests returned positive (orange bars) for Cyprus. The former is plotted on the left vertical axis while the latter on the right vertical axis.

C) In scenario C, we decrease $\beta(t)$ such that $R^{\text{model}}(t)$ approximately reaches 1 by mid August. Thus we model the case in which measures are tightened in order to preempt a second wave.

D) In scenario D, we use Eq (10) to model the effect of splitting the population into two groups. Namely, for 80% of the population we take $\beta(t)$ to remain unchanged as in scenario A, with the remaining 20% having a $\beta(t)$ that reproduces $R^{\text{model}}(t) \simeq 0.5$. This scenario models a situation in which, for example, restrictions are imposed on vulnerable groups.

**Table 1. Fit parameters obtained for $\beta(t)$ for three choices of the final day $t_f$ used in the fit, namely for day 84 (June 1$^{\text{st}}$), 114 (July 1$^{\text{st}}$), and 128 (July 15$^{\text{th}}$), with $t = 0$ being April 9$^{\text{th}}$.** Errors are quoted with the superscript and subscript for the upper and lower bounds respectively, obtained via a Markov chain Monte Carlo as explained in the text.

| $t_f$ [days] | $b_0$ [days$^{-1}$] | $b_1$ [days$^{-1}$] | $b_2$ [days$^{-1}$] | $m_1$ [days$^{-1}$] | $m_2$ [days$^{-1}$] |
|---|---|---|---|---|---|
| 84 | $0.550^{+0.006-0.001}$ | $0.019^{+0.009-0.005}$ | $0.143^{+0.009-0.009}$ | $0.082^{+0.001-0.002}$ | $0.019^{+0.002-0.004}$ |
| 114 | $0.553^{+0.004-0.001}$ | $0.015^{+0.011-0.005}$ | $0.155^{+0.005-0.011}$ | $0.080^{+0.001-0.001}$ | $0.020^{+0.002-0.003}$ |
| 128 | $0.553^{+0.005-0.001}$ | $0.014^{+0.012-0.005}$ | $0.157^{+0.005-0.001}$ | $0.080^{+0.001-0.001}$ | $0.020^{+0.002-0.004}$ |

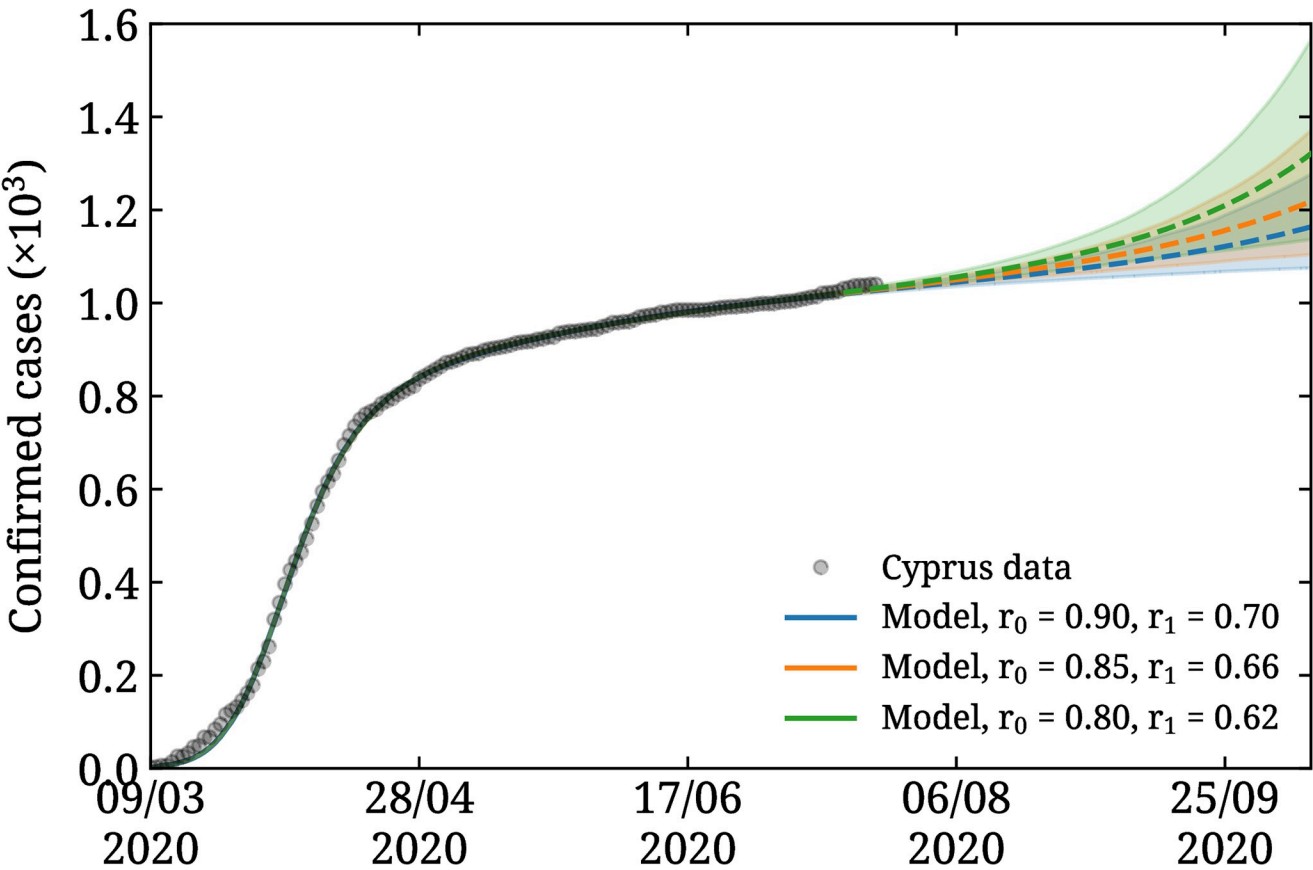

**Fig 5. Sensitivity to detection rate.** The result of fitting our SEIQR model using $(r_0, r_1) = (0.9, 0.7)$ (blue curve and band), $(0.85, 0.66)$ (orange curve and band), and $(0.80, 0.62)$ (green curve and band). In all cases the fit includes confirmed cases (black points) until $t_f = 128$ (July 15th).

These four scenarios are useful in demonstrating how we can modify features of our models to gain insights on the evolution of the pandemic, however they are idealized in that they assume smooth changes of the infection rate over long periods of time. Long-term predictions about the evolution of the disease largely depend on factors the modeling of which is beyond the scope of this work, such as policy and public response to imposed measures, availability of vaccinations, and travel. In the addendum of this paper we address how longer time series of available data can be modeled for short-term predictions, using data that have become available while this manuscript was under review.

The forecasting for the four scenarios using the SEIQR model are shown in Fig 6, when using the parameters as determined by fitting the Cyprus reported cases taking $t_f = 128$ or July 15th. As can be seen, for scenario A, the mean value of infected steadily increases. In scenarios B, C, and D on the other hand, within the uncertainty bands, we observe a flattening of the daily cases until end of the year.

Our predictions based on the time evolution of the particle model for the four different scenarios are shown in Fig 7. We use the same $r(t)$ as used in the extended SEIQR model. For the functional form of $u(t)$, we use $\beta(t)$, where now instead of $b_j$ of Eq (7) we adjust the particle velocities $u_j$ for $j = 0, 1,$ and 2 to minimize $\chi^2$ defined in the same way as in the SEIQR model given in Eq (17). The error bands in the upper part of the plots are statistical, obtained as the 90% confidence level from independently seeding the particle model 32 times. Requiring an

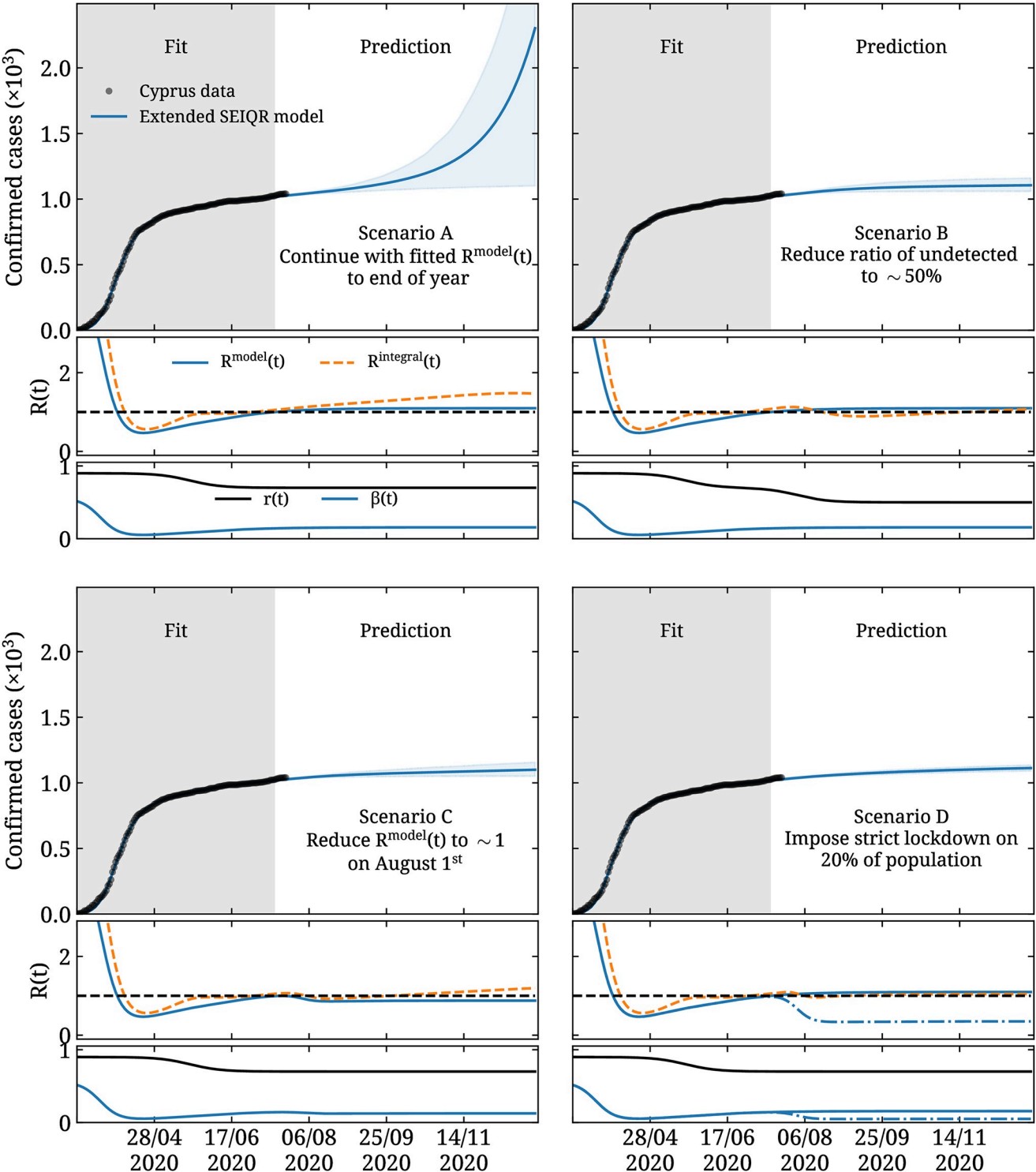

**Fig 6. Forecasting using the SEIQR model.** Predicting the evolution of COVID-19 for scenarios A and B (upper panel) and C and D (lower panel) obtained using our extended SEIQR model. In the upper part of each plot we show the reported cases in Cyprus (circles) as a function of time. The gray band shows the period used to fit the parameters, namely we use data on the reported cases up to $t_f = 128$ or July 15[th]. The blue curve and band shows our prediction for each scenario. In the central part of each plot, we show with the solid blue curve $R^{model}(t)$, and with the dashed yellow curve $R^{integral}(t)$. For the case of scenario D, the solid blue curve corresponds to $R^{model}(t)$ evaluated using $\beta(t)$ used for 80% of the population, while the dash-dotted line shows $R^{model}(t)$ evaluated using $\beta(t)$ used for 20%. In the bottom part we show $r(t)$ (black curve) and $\beta(t)$ (blue curve).

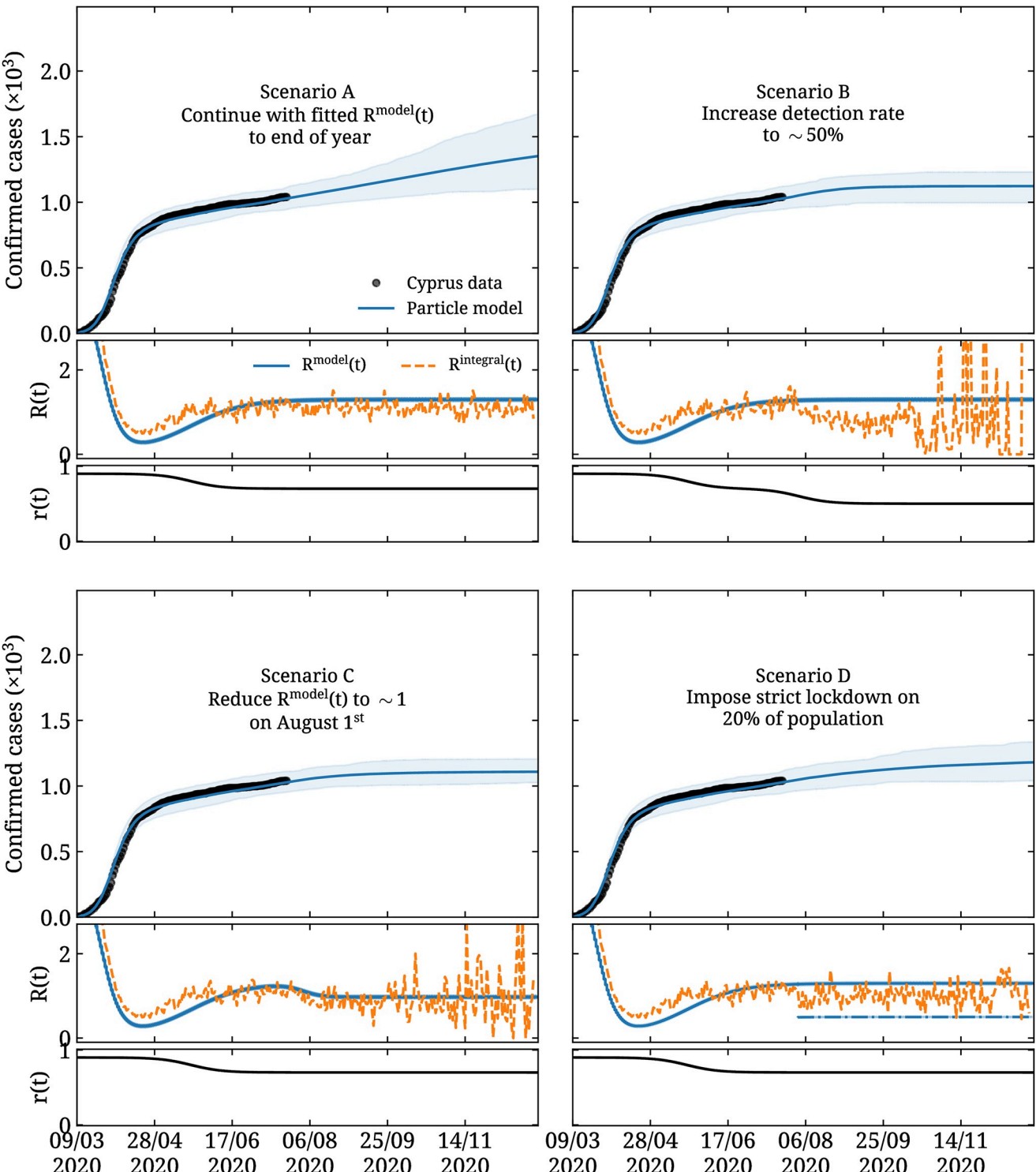

**Fig 7. Forecasting using the particle model.** Predictions for the same four scenarios as in Fig 6 but using the particle model. In the top part of the plots, the blue curve and associated band is obtained as the average and 90% confidence interval when seeding the model 32 times. In the central part of the plots, we show with the blue curve $R^{model}(t)$ and with the orange dashed curve $R^{integral}(t)$. The rest of the notation is the same as that used in Fig 6.

increase of the minimum $\chi^2$ yields errors in the parameters that are smaller than 0.5% and are therefore negligible compared to the statistical errors shown in Fig 7.

As can be seen from comparing the forecasts shown in Figs 6 and 7, the two models are qualitatively in agreement in their predictions for all four scenarios. In particular, they both predict that if infection rates remain the same as on July 15[th], within scenario A and within the uncertainty one cannot exclude either a steady future increase in daily cases that can turn into an epidemic or a flattening. Namely, for the compartmental model, the number of predicted cumulative cases for the 31[st] of Dec. has central value 2302 with 90% confidence interval yielding the range 1101 to 3556. For the particle model, the same forecast is 1352 with range 1099 to 1674.

For the other three scenarios, which assume measures are taken that will decrease infection rate and/or increase testing, both models show a flattening or complete suppression of daily cases by the end of the year. It should be noted that the large fluctuations observed in $R^{\mathrm{integral}}(t)$ in Fig 7 arise from the discrete nature of the particle model when multiple consecutive days yield zero new cases.

While this work was under review Cyprus experienced a second wave, resulting in confirmed cases rising at a larger rate than our worst case, scenario A. In the addendum we address how we model the longer term time evolution of the pandemic and compare to scenario A.

## 4 Conclusions

In this work we developed two different approaches to model the evolution of COVID-19. The two approaches can be used in any country and do not required large numbers of data. In particular they can be applied in countries like Cyprus, where the most consistent and reliable data are for the daily reported cases since data on e.g. deaths, intubations, and hospitalizations are too few for a meaningful statistical analysis. The models are highly complementary; the SEIQR model is an ODE-based compartmental approach and is computationally fast allowing sampling of parameters over long Monte Carlo Markov chains; the particle simulation approach, while computationally demanding, allows for tagging and tracing individuals and changing infection probabilities for arbitrary subsets of the population.

We calibrate and validate the models using the Cyprus reported positive COVID-19 data. For both models, we are able to fit the daily reported cases using five parameters that describe either the change of infection rate over time for the case of the SEIQR model or the velocities of the particles for the case of the particle model. In addition, we are able to obtain consistent results when using the two models to predict the evolution of COVID-19 for four scenarios that can be applied to control the epidemic.

The four scenarios are chosen as appropriate examples to demonstrate the range of parameters that can be adjusted using our two models, such as future lockdowns or easing of restrictions for subsets of the population and changes in the number of tests performed that in turn change the ratio of infected reported. Richer scenarios can be forecasted by combining these measures and by selectively applying them to multiple subsets of the population.

Although the robustness of the model fits we have undertaken pertains to datasets from other countries, we have chosen to limit the discussion to the case for Cyprus for the sake of brevity. Furthermore, future work to enhance the models will include allowing for non-uniform spatial distributions to model different population densities and data-driven modeling of the detection rate as more data become available.

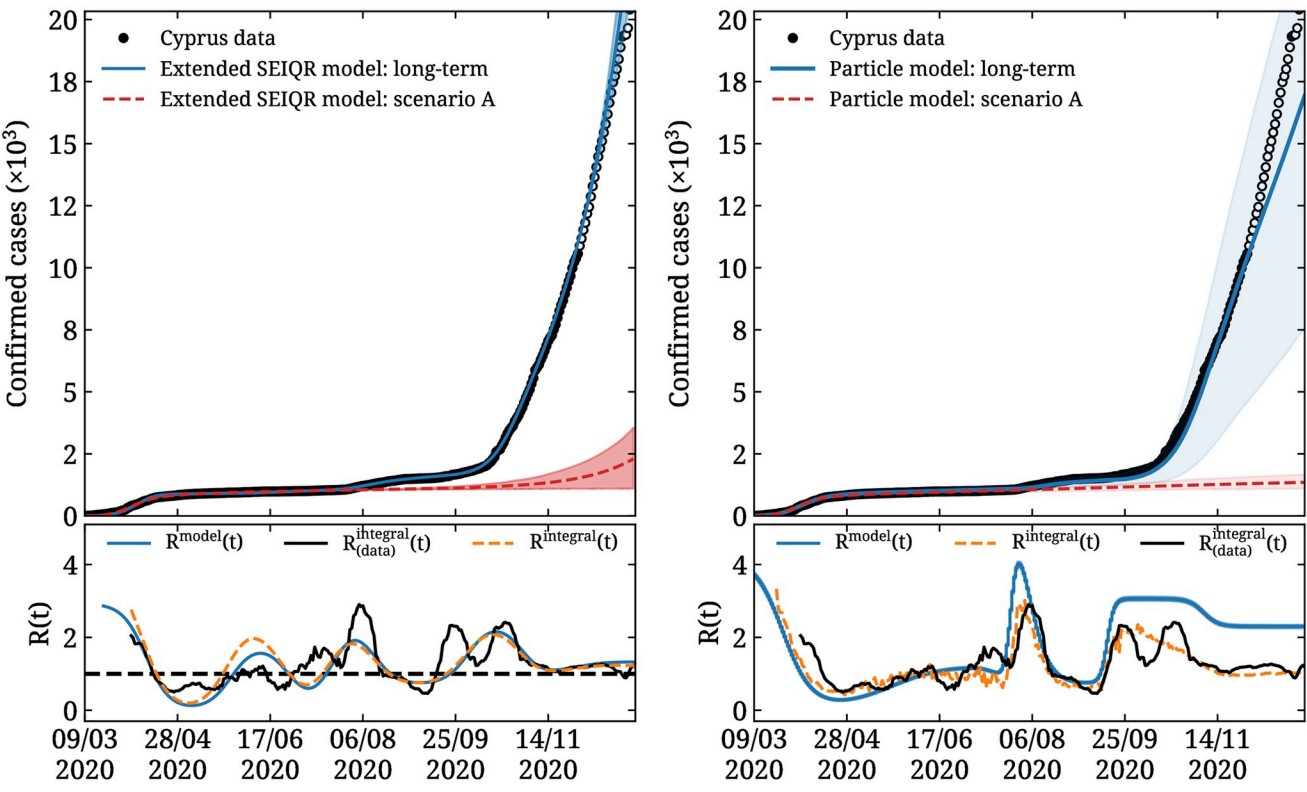

**Fig 8. Modeling of longer time series.** The SEIQR model (left) and the particle model (right) fitted to the Cyprus cumulative confirmed cases up to November 30$^{th}$ (solid blue curve and band) as explained in the addendum and compared to scenario A (dashed red curve and band). The filled black circles are the Cyprus confirmed cases included in the fit, while the open black circles are not included in the fit. The forecast is for one month, *assuming the infection rate stays constant*. In the lower panels we show the effective reproduction number as obtained by applying Eq (16) to the Cyprus confirmed cases denoted as $R^{\text{integral}}_{(\text{data})}(t)$ (black curve), in addition to $R^{\text{model}}(t)$ (blue curve) and $R^{\text{integral}}(t)$ (orange dashed curve).

## 4.1 Addendum

As mentioned in Sec. 2, for longer time series data we determine the number of inflection points $M$ and their values $t_j$ by analyzing the second derivative of the confirmed cases $c_t$. To demonstrate this approach, we carry out an analysis fitting the SEIQR model to data until November 30$^{th}$ to obtain $b_j$, $j = 0, \ldots, M$ with $M = 7$ and $m_j = 0.08$ fixed. We note that these inflection points are obtained by analyzing the confirmed cases reported and the interpretation of their causes is non-trivial and beyond the scope of our modeling, given that they depend not only on the imposition and lifting of measures but also to factors such as the response of society to the measures. The result is shown in the left panel of Fig 8, where we see that the existing data are described well by the model. Comparison with scenario A, which assumed a long-term, constant infection rate, reveals a worsening of the situation compared to the assumptions made within that scenario. In the lower panel, we show $R^{\text{model}}(t)$ and $R^{\text{integral}}(t)$, which are obtained from the modeled confirmed cases as in Figs 6 and 7. We also show $R^{\text{integral}}_{(\text{data})}(t)$, obtained by using a rolling 14 day average of the Cyprus data for the daily confirmed cases $\rho(t)$ in Eq (16). We compare these three definitions of $R(t)$ to asses whether the number of inflection points identified is consistent with the underlying data. As can be seen, all three definitions are qualitatively in agreement displaying similar main features. For the particle model, shown in the right panel of Fig 8, we similarly adjust the velocities to reproduce the daily confirmed cases $c_t$. The same comparison is made between the three definitions of $R(t)$ that are

again in qualitative agreement. In both cases, the one month forecast made *assuming the infection rate remains constant*, i.e. similar to scenario A above, yields consistent predictions for the number of confirmed cases until the end of 2020 within the two model uncertainties. This demonstrates how the models can capture the behavior of longer time series, by adjusting the parameters of both models. We are able to reproduce the data with a minimal set of parameters and both models agree in the forecasted one month trajectory of the pandemic assuming infection rates do not change.

## Acknowledgments

We would like to thank Maria Koliou Mazeri for fruitful communication and exchanges during the authoring of this paper.

## Author Contributions

**Conceptualization:** Constantia Alexandrou, Vangelis Harmandaris, Nikos Savva.

**Data curation:** Constantia Alexandrou, Anastasios Irakleous, Giannis Koutsou, Nikos Savva.

**Formal analysis:** Anastasios Irakleous, Giannis Koutsou, Nikos Savva.

**Funding acquisition:** Constantia Alexandrou, Vangelis Harmandaris, Giannis Koutsou, Nikos Savva.

**Investigation:** Vangelis Harmandaris, Anastasios Irakleous, Nikos Savva.

**Methodology:** Constantia Alexandrou, Vangelis Harmandaris, Anastasios Irakleous, Nikos Savva.

**Project administration:** Constantia Alexandrou, Vangelis Harmandaris.

**Resources:** Constantia Alexandrou, Vangelis Harmandaris, Giannis Koutsou.

**Software:** Anastasios Irakleous, Giannis Koutsou, Nikos Savva.

**Supervision:** Constantia Alexandrou, Vangelis Harmandaris, Giannis Koutsou.

**Validation:** Anastasios Irakleous, Giannis Koutsou, Nikos Savva.

**Visualization:** Constantia Alexandrou, Anastasios Irakleous, Giannis Koutsou, Nikos Savva.

**Writing – original draft:** Constantia Alexandrou, Vangelis Harmandaris, Anastasios Irakleous, Giannis Koutsou, Nikos Savva.

**Writing – review & editing:** Constantia Alexandrou, Vangelis Harmandaris, Anastasios Irakleous, Giannis Koutsou, Nikos Savva.

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
