## [Decision Letter · Decision Letter 0]

27 Jan 2021

PONE-D-20-29820

Modeling the evolution of COVID-19

PLOS ONE

Dear Dr. Koutsou,

Thank you for submitting your manuscript to PLOS ONE. After careful consideration, we feel that it has merit but does not fully meet PLOS ONE’s publication criteria as it currently stands. Therefore, we invite you to submit a revised version of the manuscript that addresses the points raised during the review process.

Please carefully follow the recommendations of both reviewers. The criticism of reviewer #1 is very strong, but you should try to overcome it by a detailed discussion in the text of the paper on all raised questions.

We look forward to receiving your revised manuscript.

Kind regards,

Vygintas Gontis, Ph.D.

Academic Editor

PLOS ONE

Journal Requirements:

Reviewers' comments:

Reviewer's Responses to Questions

**Comments to the Author**

1. Is the manuscript technically sound, and do the data support the conclusions?

Reviewer #1: No

Reviewer #2: Yes

2. Has the statistical analysis been performed appropriately and rigorously? 

Reviewer #1: Yes

Reviewer #2: Yes

3. Have the authors made all data underlying the findings in their manuscript fully available?

Reviewer #1: Yes

Reviewer #2: Yes

4. Is the manuscript presented in an intelligible fashion and written in standard English?

Reviewer #1: Yes

Reviewer #2: Yes

5. Review Comments to the Author

Reviewer #1: Please see file in attachment.

Reviewer #2: In the paper under review, the authors give a detailed study of two models: first is the

modification of classical deterministic SIR model for epidemics in the case when the transition rates between the corresponding compartments are incorporated. Second model is a particle model, in which the disease is transmitted through particle collisions and infection rates are varied by adjusting the particle velocities. Both models are applied to Cyprus data.

It is needless to say that there are thousands of papers and approaches for COVID modelling and, clearly, it is almost impossible to choose which is the right one. The authors decided to choose the simple models with a minimal number of parameters, what is rather well motivated in the case of small countries such as Cyprus. However, in general, the long-term COVID forecasting based on a mathematical model (especially deterministic SIR-type) is very questionable (see [1], [2]), especially

when making the assumptions of the model.

In the context of many COVID modelling approaches, the subject and results of the paper are interesting enough, the paper is well written and I recommend it for publishing in the journal. Below, please find

a list of comments.

6. PLOS authors have the option to publish the peer review history of their article (what does this mean?). If published, this will include your full peer review and any attached files.

Reviewer #1: No

Reviewer #2: No

---

## [Author Response · Author response to Decision Letter 0]

17 Feb 2021

Responses to all reviewer comments and suggestions are found in the attached PDF. We also attach a PDF marking the differences with the originally submitted version.

---

## [Decision Letter · Decision Letter 1]

31 Mar 2021

PONE-D-20-29820R1

Modeling the evolution of COVID-19 via compartmental and particle-based approaches: application to the Cyprus case

PLOS ONE

Dear Dr. Koutsou,

Thank you for submitting your manuscript to PLOS ONE. After careful consideration, we feel that it has merit but does not fully meet PLOS ONE’s publication criteria as it currently stands. Therefore, we invite you to submit a revised version of the manuscript that addresses the points raised during the review process.

Your revised version of the manuscript is almost acceptable for publication. Nevertheless, you should carefully take into account the recommendations of the reviewer after the second reading of your manuscript. 

We look forward to receiving your revised manuscript.

Kind regards,

Vygintas Gontis, Ph.D.

Academic Editor

PLOS ONE

Journal Requirements:

Reviewers' comments:

Reviewer's Responses to Questions

**Comments to the Author**

1. If the authors have adequately addressed your comments raised in a previous round of review and you feel that this manuscript is now acceptable for publication, you may indicate that here to bypass the “Comments to the Author” section, enter your conflict of interest statement in the “Confidential to Editor” section, and submit your "Accept" recommendation.

Reviewer #1: (No Response)

2. Is the manuscript technically sound, and do the data support the conclusions?

Reviewer #1: Partly

3. Has the statistical analysis been performed appropriately and rigorously? 

Reviewer #1: Yes

4. Have the authors made all data underlying the findings in their manuscript fully available?

Reviewer #1: Yes

5. Is the manuscript presented in an intelligible fashion and written in standard English?

Reviewer #1: Yes

6. Review Comments to the Author

Reviewer #1: I would like to thank the authors for their resubmission.

The authors have partially addressed my main concerns.

However, their replies and, more importantly, the changes they have made to the manuscript are not yet enough to grant publication. I will detail below what remains to be clarified. If the authors manage to convincingly address the following points I would be happy to recommend publication of their manuscript.

MAIN COMMENTS

(1)

Limitations to the forecasting potential of the model.

In my previous report this was the main criticism to the manuscript. The authors have partially addressed it but mostly in their reply. They must state the limitations more clearly and make them more evident in the manuscript.

For instance, concerning the statement:

“This study also demonstrates the success of these models when relatively small numbers of cases are involved and for which statistical models yield results with very large uncertainties.” I find it difficult to claim the modeling was successful. The authors must state that the predictions are reliable only for short period of times and only if the future evolution of infection rates and testing rates is specified in detail.

I still find the long-term predictions shown in the plots to be too far fetched and potentially misleading if employed for policy-making.

In order to give an intuitive measure of how restrictive the limitations of the models are, the authors should show, together with the predictions, the actual development of the confirmed case. This additional information would give a clear representation of how severe the limitations are and make it easier to appreciate that the value of these kinds of models is mostly on short term predictions.

The actual time evolution of the confirmed cases must be shown in all panels of Figure 4, 5 and 6.

If the plots become difficult to read because of the difference in magnitude between predictions and reality the authors can provide additional figures. However, I strongly recommend that in each figure both predictions and confirmed cases are always shown.

(2)

The authors state “We stress that additional scenarios can be analyzed to reflect new different circumstances as they evolve e.g. an increase of cases from incoming people, etc. The strengths of the models is that they can be

adjusted to new measures and human behavior by adjusting the reproduction number. We include an addendum with an example of how these measures can be included using data that have become available after submission of this work.”

I would recommend adding a statement to stress that this is not completely trivial and that it can become difficult to interpret what the new evolutions of the reproduction number mean. For instance, is there a possible interpretation for these 7 step-like changes that are used for fitting the additional data? How would one account for smooth changes in the reproductive number such as the ones usually due to seasonality?

The authors must comment on this in the manuscript

(3)

The authors state “We find our fits are robust to small changes r0 and r1 as long as their ratio is maintained and excluding extrema such as …” It would be good to see a more quantitative assessment of this statement, perhaps plotting what changes in the predictions using different choices of r0 and r1.

(4) The additional plots provided by the authors in reply to my former question (4) and parts of the authors’ reply should be included in the manuscript. This would make it easier to quantify their statements

(5) The reply to my former question (7) contains useful information, it should be added in the main text. Also, more importantly, the plots should not cut the values of scenario A for the compartmental model. The use of insets may help the author provide all the necessary information.

OTHER COMMENTS

(1) The authors introduced a remark that is useful to better appreciate the value of the their two, complementary, modeling approaches. However, the new statement “This leads to confidence on the model forecasts on one hand, but also allows us to attribute different errors to the predictions, since the particle model exhibits stochastic errors, related to how the pandemic was seeded within the population, while the compartmental model exhibits modeling uncertainty, based on the uncertainties of the fitted parameters." requires some clarification.

It seems like the stochastic models require parameter fitting as well. It would be more precise to say that their uncertainty is due both to stochasticity and uncertainty in parameters.

7. PLOS authors have the option to publish the peer review history of their article (what does this mean?). If published, this will include your full peer review and any attached files.

Reviewer #1: No

---

## [Author Response · Author response to Decision Letter 1]

9 Apr 2021

We thank the reviewer for their comments. We have revised the manuscript and respond to each of the reviewer's comments in the attached response.

---

## [Editor Report · Decision Letter 2]

13 Apr 2021

Modeling the evolution of COVID-19 via compartmental and particle-based approaches: application to the Cyprus case

PONE-D-20-29820R2

Dear Dr. Koutsou,

We’re pleased to inform you that your manuscript has been judged scientifically suitable for publication and will be formally accepted for publication once it meets all outstanding technical requirements.

Kind regards,

Vygintas Gontis, Ph.D.

Academic Editor

PLOS ONE
---

## [Editor Report · Acceptance letter]

26 Apr 2021

PONE-D-20-29820R2 

Modeling the evolution of COVID-19 via compartmental and particle-based approaches: application to the Cyprus case 

Dear Dr. Koutsou:

I'm pleased to inform you that your manuscript has been deemed suitable for publication in PLOS ONE. Congratulations! Your manuscript is now with our production department. 

Kind regards, 

on behalf of

Dr. Vygintas Gontis 

Academic Editor

PLOS ONE